

## The role of light as vital effect on coral skeleton oxygen isotopic ratio

Anne Juillet-Leclerc

LSCE Domaine du CNRS, 91198 Gif-sur-Yvette, France

*Correspondence to*: Anne.Juillet-Leclerc@lsce.ipsl.fr

## Abstract

Light, an environmental parameter playing a crucial role in coral aragonite growth and $\delta^{18}O$ formulation, is always neglected in the geochemical literature. However, by revisiting already published studies, we demonstrated that light might be considered as a vital effect affecting coral aragonite oxygen isotopic ratios.

Re-examining data series included in a publication by Weber and Woodhead (1972), we stressed that annual $\delta^{18}O$–annual temperature calibrations of all considered coral genera may be compared because their assessment assumes homogenous light levels. Temperature prevails on $\delta^{18}O$ because it influences $\delta^{18}O$ in two ways: firstly it acts as is thermodynamically predicted implying a $\delta^{18}O$ decrease; and secondly it induces an enhancement of photosynthesis causing $\delta^{18}O$ increase. When the highest annual temperature occurs simultaneously with the highest annual irradiation, the annual $\delta^{18}O$ amplitude is shortened. The annual $\delta^{18}O$–annual temperature calibration is also explained by the relative distribution of microstructures, centres of calcification or COC and fibers, according to morphology, and in turn taxonomy. We also investigated monthly $\delta^{18}O$–monthly temperature calibrations of *Porites* grown at the same sites as by Stephans and Quinn (2002), Linsley et al. (1999, 2000) and Maier et al. (2004). Multiple evidence showed that temperature is the prevailing environment forcing on $\delta^{18}O$ and that the mixture of temperature and light also determines the relative distribution of microstructures,



explaining the relationships between *Porites* calibration constants. By examining monthly and annual
$\delta^{18}O$–monthly and annual temperature calibrations, we revealed that monthly calibration results from
the superimposition of seasonal and annual variability over time. Seasonal $\delta^{18}O$ strongly impacted by
seasonal light fluctuations, may be obtained by removing interannual $\delta^{18}O$ only weakly affected by
light. Such features necessitate the reconstitution of tools frequently utilised, such as the coupled
$\delta^{18}O$–Sr/Ca or pseudo-coral concepts.

## 34  1 Introduction

The oxygen isotope data preserved in the scleractinian coral skeleton is an excellent proxy for
temperature and/or $\delta^{18}O_{seawater}$ variability (McConnaughey, 1989), following the concept of isotopic
thermometer (Urey, 1947). However, coral aragonite $\delta^{18}O$ is depleted relative to the isotopic values of
the ambient seawater (Weber and Woodhead, 1972), inducing anomalies commonly termed as vital
effects. Therefore, we need to identify environmental parameters really included in the $\delta^{18}O$ time-
series, the most-used coral skeleton proxy so far.
Because temperature and light intensities are always strongly related in the field, the real impact of
light on $\delta^{18}O$ cannot be decoupled from the temperature effect. Although the relationship between
light and calcification has long been recognised (Goreau, 1959; Gattuso et al., 1999), the control of
ambient light level on the isotopic disequilibrium offset of coral aragonite from seawater could only be
speculated upon (Land et al., 1975; McConnaughey, 1989; Felis, 2003). The role of light on coral $\delta^{18}O$
can only be proved from evidence provided by cultured corals in controlled light conditions, all the
other parameters remaining constant (Reynaud-Vaganay et al., 2001; Juillet-Leclerc and Reynaud,
2010). The latter authors show that at 25°C, $\delta^{18}O$ measured on *Acropora* clearly increases due to
photosynthesis enhancement accompanying raised light intensity, while the skeleton exhibits
noticeable infilling, accompanied by a reduced linear extension (Juillet-Leclerc and Reynaud, 2010).
In addition, temperature increase is also responsible for photosynthesis enhancement (Juillet-Leclerc
et al., 2014). The comparison of biological measurements, such as net productivity and zooxanthellae
density (Juillet-Leclerc et al., 2014) highlights the evidence, well known by biologists, that symbiont



distribution on a coral is not homogeneous, varying with coral genera and coral morphology, ambient
vegetation in water column, water column depth and potential adaptations (Porter et al., 1984; Kühl et
al., 1995; Karako-Lampert et al., 2004; Iluz and Dubinsky, 2015). Juillet-Leclerc et al. (2014) deduced
that light should impact each temperature calibration and should be, likely to a large extent,
responsible for the vital effect.
In the field, according to location, temperature through the seasonality of precipitation and, in turn,
nebulosity, is positively or negatively correlated to irradiation. Therefore, seasonal isotopic amplitude
should not only reflect the temperature but the global effect of both temperature and light.
Additionally, colonies collected on corals grown in the same location do not receive equivalent
irradiation according to the water depth and their environment (Felis et al., 2003) and/or samples are
not influenced by similar zooxanthellae density due to coral morphology (Land et al., 1974; Juillet-
Leclerc, 2014). Consequently, we supposed that $\delta^{18}$O–temperature calibrations established from
seasonal isotopic data are strongly impacted by local seasonality and/or characteristics of each colony.
But calibrations based on annual data provided by one coral should not be similarly impacted by light
as the seasonal data calculated from monthly samples. In turn, the pure temperature impact on $\delta^{18}$O
cannot be quantified. Considering that light impacts coral $\delta^{18}$O, what level of $\delta^{18}O_{Seawater}$ may be
included in $\delta^{18}$O determination? Is there a hierarchy between temperature and light influence on $\delta^{18}$O?
We intend to illustrate the different light effects on coral isotopic calibration *versus* temperature by
using earlier published evidence.
The paper is structured as follows. First, we will revisit the dataset of Weber and Woodhead (1972),
characterised by the unique sampling mode allowing the comparison of several annual coral genera
calibrations. Second, we will compare seasonal calibrations estimated for several *Porites* colonies
collected in warm and mediate-temperature water (Linsley et al., 1999; Maier et al., 2004; Quinn and
Sampson, 2002). Third, we will show that seasonal and annual $\delta^{18}$O–temperature calibrations are not
linearly related (Crowley et al., 1999; Boiseau et al., 1998). Then, understanding that the entanglement
of environmental parameters and potential tracers captured in coral skeleton over the time imposes the
use of statistical multi-proxy treatment as proposed by Hugues and Amman (2009), we will discuss



the ways to circumvent what is hidden behind the blurred term 'vital effect' (Lowenstam and Weiner,

1989).


## 2 Weber and Woodhead (1972) paper revisited

### 2.1. Data series

Data published by Weber and Woodhead (1972), in the following referred to as WW72, remains one
of the most exhaustive coral $\delta^{18}O$ databases, although all the compiled data are not available in the
publication. Our present knowledge may shed new light on the coral $\delta^{18}O$–temperature dependence.
As early as 1951, Urey suspected that physiological processes could affect the $\delta^{18}O$ of calcareous
organisms, leading to values out of isotopic equilibrium with seawater, as shown by $\delta^{18}O$ and $\delta^{13}C$
measured in corals collected from Heron Island (Australia) (Weber and Woodhead, 1970). However,
despite depleted $\delta^{18}O$, these values showed apparent temperature dependence. In order to verify the
coral skeleton's capability to capture temperature, they collected several coral genera spread over
tropical oceans (Fig. 1a). WW72 data were used to establish a formula able to predict past SST
following the isotopic thermometer concept (Urey 1947), expressed as:

$$SST°C = A + B \times \delta^{18}O \ (‰) \qquad Eq. \ (1)$$

Annual temperature distribution from 21.2 to 29.3°C was prescribed by the 29 sites spread in the
Pacific Ocean (except for two sites in the Atlantic and Indian oceans) (Fig. 1a). Several specimens of
all the genera present on a site, at most 44 coral genera were collected; for example, 54 specimens of
*Acropora* were collected on Heron Island (Great Barrier Reef) or 39 *Porites* in the Torres Strait
(between Queensland and Papua New Guinea). Because *Acropora* and *Porites* are ubiquitous genera,
*Acropora* $\delta^{18}O$ calibration was derived from 835 samples and 421 samples of *Porites*. Derived
calibrations (Fig. 1b) may be considered as statistically significant. In addition, isotopic analyses were
conducted on annual samples, identified by X-ray growth bands, a pair of clear and dark bands
corresponding to the annual growth (Barnes and Lough, 1996).

*2.1.1 WW72 calibrations*



Since the goal of the WW72 study was to verify the relationship between $\delta^{18}$O and SST to predict SST
(Epstein 1951, 1953), they established the calibration of temperature relative to $\delta^{18}$O: PT =
'paleotemperature curve' i.e. the temperature expressed according to $\delta^{18}O_{carbonate}$ or as reported in
Table 4 from the WW72 formula Eq. (1) (Fig. 1b). In this study, the authors neglected the term
$\delta^{18}O_{seawater}$. However, in 1972, the annual instrumental temperature precision was much greater than
$\delta^{18}O_{carbonate}$ measurement precision. To date, our purpose is not to prove the existence of an isotopic
thermometer but rather to check the reliability of the relationship between δ18Ocarbonate and SST.
Therefore, we calculated the relationship:
$$\delta^{18}O_{carbonate} = a \times SST\ (°C) + b \qquad \text{Eq. (2)}$$
where a and b are constants (Fig. 1c), considering that the variable is SST.
We are aware that by inverting (1) into (2), from the same dataset, the obtained relationship has not
equivalent significance and similar errors in the constants than the relationship calculated from
initially published calibrations.

*2.1.2 Calibrations including $\delta^{18}O_{seawater}$*

The WW72 dataset does not take into account δ18Oseawater. Juillet-Leclerc and Schmidt (2001) included
annual $\delta^{18}O_{SeaWater}$ values assessed in the calibration established for *Porites* following the formula:
$$\delta^{18}O_{carbonate} - \delta^{18}O_{seawater} = \alpha \times SST(°C) + \beta \qquad \text{Eq. (3)}$$
where α and β are constants. They obtained:
$$\delta^{18}O_{carbonate} - \delta^{18}O_{seawater} = -0.20 \times SST(°C) + 0.45 \qquad \text{Eq. (4)}$$
with $R^2 = 0.83$, N = 22, p < 0.001, only significant over the SST range from 24 to 30 °C (Juillet-
Leclerc and Schmidt, 2001), by introducing δ18Oseawater following two estimates (Fig. 2). The
correlation linking $\delta^{18}$O directly to temperature showed a higher coefficient:
$$\delta^{18}O_{carbonate} = -0.27 \times SST(°C) + 2.24 \qquad \text{Eq. (5)}$$
with $R^2 = 0.91$, N = 24, p < 0.001 (Fig. 1c) including the lowest temperatures neglected in Eq. (4)
(Juillet-Leclerc and Schmidt, 2001).





A similar procedure was conducted for *Acropora*, using the same $\delta^{18}O_{seawater}$ estimates. We obtained:
$$\delta^{18}O_{carbonate} - \delta^{18}O_{seawater} = -0.21 \times SST\ (°C) + 1.26 \qquad\qquad \text{Eq. (6)}$$
with $R^2 = 0.87$, $N = 24$, $p < 0.001$ significant over the temperature range from 21 to 30 °C (Fig. 2). In
contrast, *Porites* isotopic values associated to the lowest temperatures are included in the calibration.
The correlation linking $\delta^{18}O$ and temperature showed a higher coefficient:
$$\delta^{18}O_{carbonate} = -0.28 \times SST(°C) + 3.36 \qquad\qquad \text{Eq. (7)}$$
with $R^2 = 0.97$, $N = 24$, $p < 0.001$ (Fig. 2).
Slopes (a) shown by *Porites* and *Acropora* temperature calibrations including $\delta^{18}O_{seawater}$, –0.20 and –
0.21‰/°C respectively differ from those deriving only from $\delta^{18}O_{carbonate}$ and temperature. They are
close to the slope of –0.19 ‰/°C assessed for inorganic aragonite calibration (Kim et al., 2007).
Slopes have been obtained from other genera such as *Platygira*, *Montipora* or *Pavona* (Fig. 2) in
different temperature ranges and with variable correlation coefficients. The number of analysed
specimens is reduced compared to *Acropora* or *Porites* calibrations (WW72).

*2.1.3 Relationship between a and b*

When comparing constants (a) and (b) from equation (2) for all the genera annual $\delta^{18}O$ *versus* annual
temperature of theWW72 data series, we obtained a strongly significant linear relationship:
$$b = -27.9a - 5.13 \qquad\qquad \text{Eq. (8)}$$
with $R^2 = 0.95$, $N = 29$ and $p < 0.001$. We verified that after neglecting extreme values of (b), the
relationship remained significant ($R^2 = 0.90$, $N = 26$ and $p < 0.001$) (Fig. 3a). Such a relationship is not
hazardous, but reflects inherent features of annual coral $\delta^{18}O$–annual temperature calibrations.
We observed in Fig. 1c that some curves converged, defining several bundles. All the groups, formed
by genera gathered in the same bundle, are listed in Table 1. In Fig. 1d, we underline that, for example
for *Acropora* and *Porites* groups, the convergence corresponds to quantified temperature and isotopic
value ranges. When comparing constants of calibrations corresponding to a group, we obtained linear
relationships, all showing highly significant correlation coefficients: $R^2 = 0.99$ (Fig. 3b).





**2.2 Improved meaning of annual calibration from WW72**
Each temperature value, corresponding to one island, is associated to the averaged $\delta^{18}O$ measured for
corals of the same species, all receiving identical local irradiation. However, several colonies of a
same genus might be subjected to different light incidence, intensity depending to relative growth
depth, or corals included in the same local environmental could have morphological portion
containing higher or lower zooxanthellae distribution or potential adaptation (Porter et al., 1984; Kühl
et al., 1995; Karako-Lampert et al., 2004; Iluz and Dubinsky, 2015). WW72 data corresponding to
each temperature correspond to colonies numerous enough to represent a quasi-homogenous
irradiation. This explains that calibrations assessed for all genera may be significantly compared
without taking into account light conditions (Fig. 1). Only these conditions allow the comparison of
calibrations assessed for several coral genera.

*2.2.1 Temperature dependence of coral $\delta^{18}O$*

2.2.1.1 Temperature recorded at least twice
$\delta^{18}O$ temperature dependence expressed as Eq. (2) stresses the strong temperature effect on isotopic
fractionation but this formula excludes $\delta^{18}O_{seawater}$ displayed in the classical thermodynamic expression
Eq. (6). After introducing $\delta^{18}O_{seawater}$ into WW72 data, for some genera, *Porites* and *Acropora* genera
(Fig. 2)*,* we observed that the usual thermodynamic equation is also significant but to a lower degree;
for example by taking into account only temperature, $R^2 = 0.91$ and $0.98$ instead of $R^2 = 0.87$ and $0.93$
for *Porites* and *Acropora* respectively, for the usual thermodynamic equation.
In the calibrations depending only on temperature, temperature may act first, according to
thermodynamic law (Epstein et al., 1953; Juillet-Leclerc et al., 2014) and second, through the
photosynthetic process (Juillet-Leclerc and Reynaud, 2010), which is enhanced by a temperature
increase. However, an increase in temperature induces a decrease in $\delta^{18}O$ following the first process
while the second mechanism causes a rise in $\delta^{18}O$ confusing the global isotopic effect. Temperature
influences $\delta^{18}O$ twice, explaining that temperature is the main factor on isotopic value determination,
which does not exclude the role of $\delta^{18}O_{seawater}$.




2.2.1.2 Annual coral $\delta^{18}O$ contains $\delta^{18}O_{seawater}$
Calibrations taking into account $\delta^{18}O_{seawater}$ exhibit a slope value close to that calculated for isotopic
equilibrium of inorganic aragonite with water, suggesting that under quasi-uniform light, the isotopic
offset of coral $\delta^{18}O$ (the difference between coral $\delta^{18}O$ and value at isotopic equilibrium) is constant,
regardless of temperature (Fig. 2).
Calibration deduced for *Porites* is restricted to temperatures higher than 25°C, $\delta^{18}O$ corresponding to
lowest temperatures being too high to be included on the strongly significant linear curve. For the
other genera, n is limited by the lack of these coral colonies on numerous islands.
Equations (4) and (6) confirm that, to a lesser degree than temperature, $\delta^{18}O_{seawater}$ may be included
in annual coral skeleton $\delta^{18}O$.

200   *2.2.2 Relationship between constants a and b*

WW72 data reveal a strong relationship between annual $\delta^{18}O$–annual temperature calibrations and
taxonomy because each genus shows a unique relationship. Calibration bundles defining groups
(Table 1) enhance this feature. Coral genera classification or taxonomy is based on coral morphology.
Land et al. (1975) stressed the high $\delta^{18}O$ variability following the longitudinal section on the calices of
*Eusmilia fastigiata* or the septa dentations of *Scolymia cubensis* inducing coral skeleton isotopic
variations. The authors observed that according to coral location some skeleton portions might be
more or less developed, implying a large isotopic variability.

Considering the relationship $\delta^{18}O_{carbonate} = a \times SST (°C) + b$ Eq. (2) derived from the WW72 dataset,
(a), the slope value, varies from –0.16 to –0.36. This corresponds to a disequilibrium indicator
compared to –0.19, the slope value derived from the theoretical $\delta^{18}O$–temperature equation at
equilibrium (Kim et al., 2007). The equation (7) exhibits that the constants (a) and (b) of annual $\delta^{18}O$–
annual temperature calibrations established for several annual samples collected from all WW72 coral
genera, obey to a linear relationship (Fig. 3): $b = –29.07 \times a – 5.13$ Eq. (8) with N = 37 and R² = 0.95,





p<0.001. This suggests that the temperature dependence of isotopic oxygen ratio is based on a unique
rationale according to taxonomy, inherent to the coral skeleton.

2.2.2.1 Common feature of $\delta^{18}O$ and Sr/Ca calibrations
A similar relationship exists between the constants of annual coral skeleton Sr/Ca–annual temperature
calibration (Marshall and McCulloch, 2002; Wei et al., 2007; Deng et al., 2014; D'Olivio et al., 2018),
another temperature tracer present in the coral skeleton. There is no straightforward link between $\delta^{18}O$,
oxygen being a component of CaCO3 and Sr, an impurity included in the skeletal aragonite. However,
it is possible to recognise common $\delta^{18}O$ and Sr/Ca behaviour relative to their crystalline unit
distribution in the coral skeleton and the concept of taxonomy.

2.2.2.2 Role of crystalline microstructures
It is admitted that the coral skeleton presents composite mineral microstructures: centres of
calcification (COC) and fibres, embedded in a few organic matter as a network (Von Euw et al.,
2017). These crystalline elements are distributed differently according to morphology (Cuif and
Dauphin, 1978, 2005; Stolarski, 2003; Nothdurft and Webb, 2005). The latter authors showed that
each microstructure is preferentially present in some morphological parts, which are more or less
developed following the genus. On one hand, $\delta^{18}O$ signature differs according to the microstructure
unit (Rollion-Bard et al., 2003; Blamart et al., 2005; Meibom et al., 2006; Juillet-Leclerc et al., 2009),
the COC $\delta^{18}O$ value being lower than that of the fibre. On the other hand, Sr/Ca ratios measured on
COCs are higher than those of fibres (Meibom et al., 2006; Cohen et al., 2001). Meibom et al. (2006)
sampled *Colpophyllia sp.* following microstructures on a skeleton morphology fragment and their
Sr/Ca data of each crystal type show convergence. Cohen et al. (2001) examined synchronously
deposited microstructures on *Porites lutea* over a year, exhibiting COC elemental ratios systematically
higher compared to those of fibres developed over an identical period. Thus, the annual COC Sr/Ca
value is higher than the annual fibre Sr/Ca signature (Cohen et al., 2001). Therefore, we suggest that
discrepancies of morphology existing between coral genera are due to differences of microstructure





proportions. Therefore, differences in geochemical values due to the relative number of
microstructures could explain common features between the annual trace element ratio and annual
$\delta^{18}$O–annual temperature calibrations, especially the linear relationship linking the respective
calibration constants.

*2.2.3 Notion of optimal growth*

We already identified groups of genera (Table 1) showing constants (a) and (b) from calibrations of
genera linked by strong correlation coefficient ($R^2 = 0.99$) (Fig. 3b). This could be due to morphology
similarities of the genera of each bundle, characterised by identical proportions of microstructures in
each coral group skeleton. We previously highlighted the intersection of calibrations defined by
coupled $SST_{intersection}$ and $\delta^{18}O_{intersection}$, independently to $\delta^{18}O_{seawater}$ (Fig. 3b). Such a common isotopic
composition can be obtained because light intensity is homogenised. Coupled values ($SST_{intersection}$ and
$\delta^{18}O_{intersection}$) might be related to the concept of the optimal growth conditions (Jokiel and Coles, 1977)
attributed to an optimal growth temperature.
An optimal temperature, between 25 and 29°C, corresponding to an optimal growth rate has been
attributed to some coral genera, *Montipora verrucosa*, *Pocillopora damicormis*, *Fungia scutaria*
(Jokiel and Coles, 1977). However, it was difficult to identify a temperature value corresponding to
optimal growth conditions: is maximal extension rate or density considered as representative of
optimal growth rate (Carricart–Ganivet et al., 2004; Worum et al., 2007; Brachert at al., 2013)? This
concept could also be linked to the temperature corresponding to the maximal $Ca^{2+}$ input in the coral
skeleton, as has been identified by Al-Horani (2005). Optimal growth temperature differs according to
coral genus (Buddemeier and Kinzie III, 1976; Marshall and Clode, 2004). The latter authors relate the
temperature dependence of the optimal growth rate of *Galaxea facsicularis* to an enzyme-catalysed
reaction, but they finally conclude that the response of calcification rate to temperature being similar
in zooxanthellate and azooxanthellate corals, the responsible mechanism should be based on another
fundamental process. After studying the calcification of 38 *Porites* colonies, Cooper et al. (2008)



suggest that 26.7°C could be the thermal optimum of calcification rate for this genus, which could be
compared with our $SST_{intersection}$ identified for *Porites*, of 28.4°C.
From our evidence, $SST_{intersection}$ corresponds to $\delta^{18}O_{intersection}$ shared by a coral group and is related to
morphology and growth rate (Land et al., 1975), likely related to optimal growth. Assuming that
relative amounts of crystalline units are responsible for the constant relationship of the annual $\delta^{18}O$–
annual temperature calibration, we have to assume that at this temperature, identical $\delta^{18}O$ is due to the
same relative crystalline amounts in the coral skeleton, whatever is the considered genus belonging to
the same group (Table 1) or, more probably, a temperature range corresponding to an isotopic range
(Fig. 1d). For example, the coupled $SST_{intersection}$ and $\delta^{18}O_{intersection}$ could represent common values
shared by all *Acropora* or *Porites* colonies whatever is the site where they grow (Fig. 3).

The following conclusions concern all coral genera studied in WW72 and their annual $\delta^{18}O$–
calibrations. Temperature prevails on $\delta^{18}O$ because it influences $\delta^{18}O$ in two ways, first it acts as is
thermodynamically predicted, implying a $\delta^{18}O$ decrease and second it induces an enhancement of
photosynthesis causing a $\delta^{18}O$ increase. Similar behaviour of the constants of the annual $\delta^{18}O$ and
Sr/Ca–annual temperature calibrations should be explained by the presence of two crystallographic
components of the coral skeleton, showing specific COC–to–fibre proportions for each genus,
depending on their morphology and characterised by their respective geochemical signatures.
We deduce from WW72 data that all the coral genera are potential temperature tracers.
It is possible to associate to each genus (likely to each bundle) a temperature range coupled to a $\delta^{18}O$
range corresponding to optimal growth rate.

**3 *Porites* monthly calibration**
The first seasonal $\delta^{18}O$ records were measured for *Montastrea annularis* (Fairbanks and Dodge,
1979). Seasonal $\delta^{18}O$ profiles from *Porites* grown in the Galapagos (McConnaughey, 1989) were used





to assess seasonal $\delta^{18}$O–seasonal temperature calibration. Presently, such a regression is commonly
calculated.
The preliminary step of climatic reconstruction using *Porites* skeleton, the genus more often analysed
in this context, consists of the assessment of seasonal $\delta^{18}$O–seasonal temperature calibration based on
monthly instrumental temperatures over the last decades covered by the core. Sampling is conducted
along the coral's growth through time, following the maximal growth rate perpendicular to the annual
density bands shown by X-ray (DeLong et al., 2013).
In order to test seasonal $\delta^{18}$O–seasonal temperature calibration variability including the seasonal light
effect, calculated for several coral cores collected on a given site, at different temperature ranges, we
considered studies conducted on several *Porites* colonies from three sites. The mean annual
temperature offshore Amédée Island, New Caledonia (22° 29' S, 166° 28' E) was 24.72°C, over the
period 1968–1992 (Quinn and Sampson, 2002; Stephans et al., 2004), while at Clipperton Atoll (10°
18' N, 109° 13' W) the mean annual temperature was 28.5°C, over the period 1985–1995 (Linsley et
al., 1999, 2000) and in the Flores Sea, Indonesia (6° 32' S, 121° 13' E) the mean annual temperature
was above 28°C, over the period 1979–1985 (Maier et al., 2004).

**3.1 Data in the three sites**
*3.1.1 Calibrations from New Caledonia data*
Calibrations have been calculated for two paths of a long core 92 and 99-PAA and two short cores 92-
PAC and 92-PAD, collected from *Porites lutea*, compared to the appropriate grid square GISST2
temperature from 1968 to 1992 (Quinn and Sampson, 2002; Stephans et al., 2004). These data are
available on https://www.ncdc.noaa.gov/paleo/study/1877. Seasonal temperature varied from 21.5 to
27.5°C, values lower than the $SST_{intersection}$ estimated for *Porites*, of 28.4°C. Precipitation did not show
any preferential seasonality. The $\delta^{18}$O record from all the cores displayed a clear seasonal cycle
(Quinn and Sampson, 2002; Stephans et al., 2004). All the calibrations given following Eq. (2)
showed higher slope than –0.19, the slope value derived from the theoretical $\delta^{18}$O–
temperature relationship at equilibrium (Kim et al., 2007), and varying, comprised between –0.13 and



–0.17 (Fig. 4a, 4b). When comparing (a) and (b) from the calibrations, a strongly correlated
relationship is obtained (Table 2) (Fig. 4c).

*3.1.2 Calibration from Clipperton Atoll data*
We considered three *Porites* cores (Linsley et al., 1999; 2000) whose data are provided on
https://www.ncdc.noaa.gov/paleo/study/1846. Over the last decade, annual temperature varied less
than 2 °C (Reynolds and Smith, 1994), showing a mean value of 28.5 °C and a clear seasonal cycle
(Linsley et al., 1999; 2000). Maximum lag between $\delta^{18}O$ and temperature is at least 1 month,
occasionally up to 2 months (Linsley et al., 1999; 2000).
Expressed following Eq. (2), calibrations showed low slopes, compared to –0.19, the slope value
derived from the theoretical $\delta^{18}O$–temperature relationship at equilibrium (Kim et al., 2007), varying
between –0.4 and –0.53 ‰/°C (Fig. 5a, 5b), (a) and (b) being strongly correlated (Table 2) (Fig. 5c).

*3.1.3 Calibration from the Flores Sea*
Twelve pathways collected on six coral heads from three *Porites* species (*Porites lutea*, *Porites*
*murrayensis* and *Porites australiensis*) (Fig. 6a, 6b) provide 12 calibrations given following equation
(2) covering 55 months and converted into Eq. (3) (Maier et al, 2004). In the site located at the
western margin of the Warm Pool, the mean annual temperature is 28°C with an annual amplitude of
2.5°C. Although the assessed constants are known to be not free of errors, the relationship established
between (a) and (b) showed a highly significant correlation coefficient (Table 2) (Fig. 6c, 6d). It is
noticeable that several *Porites* species were considered.
We display together all the *Porites* calibrations previously mentioned in Fig. 7a and equation
corresponding to *Porites* group V from WW72, covering high temperature amplitude and
disequilibrium indicator range. The values of the constants (a) and (b) of all the calibrations are
reported in Fig. 7b. The correlation coefficient of the linear regression is 0.999, N = 25.





**3.2 Significance of the constants of δ¹⁸O-temperature calibrations derived from**
**monthly data.**
We assume that calibrations measured on different coral colonies grown at a given site (New
Caledonia, Clipperton or Indonesia) differ according to various light sensitivities due to depth or light
incidence or acclimation (Fig. 7a) because seasonality strongly affects light variations, and is likely to
be different following site location. However, calibration constants calculated from monthly data for
*Porites* remain strongly correlated (Fig. 7b) as we observed for annual δ¹⁸O–annual temperature
calibrations (2.2.2).

*3.2.1 Local effects on δ¹⁸O*
The isotopic characteristics may be explained by local conditions. In New Caledonia the mean annual
temperature, 24.72°C, is lower than the temperature intersection estimated for *Porites* group from
WW72 (Fig. 4) of 28.4°C, the annual amplitude being 6°C. To justify the weak slope of the
calibrations, we argue that maximal annual temperature and high light are synchronous: thus, the δ¹⁸O
decrease due to temperature being reduced during boreal summer and during winter is normal.
Therefore, the annual isotopic amplitude is limited. 92PAC and 99PAA show strong attenuation in
boreal summer (Fig. 4a), which could be related to strong photosynthetic activity. However these coral
cores also exhibit low δ¹⁸O during winter according to lower slopes of calibrations for 92PAC and
99PAA compared to 92PAC and 92PAD (Fig. 4b).
However, in Clipperton and in Indonesia, the mean annual temperature is about 28°C with a weak
annual temperature amplitude (about 2°C). In these conditions, the disequilibrium indicator (a) varies
from –0.6 to –0.4 (Fig. 7b). The temperature range in Clipperton and the Flores Sea is close to the
temperature intersection estimated for *Porites* group from WW72, at 28.4°C (Fig. 5 and Fig. 6
respectively). In Indonesia, slope (a) shows much higher range, from –0.4 to –1 than at Clipperton.
Maier et al. (2004) stress that calibrations are calculated from several *Porites* species (Fig. 6). The
authors also observe negative correlation between mean annual coral δ¹⁸O and annual linear skeletal
extension.






*3.2.2 Correlation of the constants derived for monthly $\delta^{18}O$–temperature calibrations*

Relationships calculated from monthly data measured in Indonesia (Fig. 6) and Clipperton (Fig. 5)

corals (Maier et al., 2004; Linsley et al., 1999) are almost the same. This could be due to the identical

temperature range (from 26 to 29°C). As calibrations do not obey only thermodynamic rules,

$\delta^{18}O_{seawater}$ is neglected. The relationships linking (a) and (b) do not depend on local environmental

parameters and seem to be inherent to *Porites* calcification, as we noticed for annual calibrations

(2.2.1.1). Furthermore, calibration constants deduced from New Caledonia corals, subject to

drastically different external conditions from in the other sites, follow the same linear relationship

(Table 2) (Fig. 7a, b). Moreover, the constants calculated for annual data of *Porites* derived from

WW72 are included in the linear relationship (Fig. 7b). The relationship b = –27.24 and a –4.92

(established with N = 19, $R^2$ = 0.999) Eq. (7) (Fig. 7b) reflects *Porites* skeleton crystallisation,

regardless of other external conditions, including light.

However, we demonstrated that light affected cultured *Acropora* (Juillet-Leclerc and Reynaud, 2010).

Is such a behaviour only restricted to *Acropora*? The latter authors attributed this feature to the

existence of two distinct crystallisation modes of COC and fibres, which are common to other

*Acropora* species (Gladfelter, 1982) but also to other genera (Jell, 1974).

Therefore, constants of monthly $\delta^{18}O$–monthly temperature calibrations show a strong relationship

(Fig. 7) due to crystalline distribution of the coral skeleton, COCs being fusiform crystals deposited

according to temperature, regardless of light intensity, ensuring linear extension whereas fibres

formation ensuring infilling is light- and temperature-dependent (Gladfelter, 1982; Juillet-Leclerc and

Reynaud, 2010; Juillet-Leclerc et al., 2018). We have already highlighted that the relationship linking

constants of annual $\delta^{18}O$–annual temperature calibration is due to the relative numbers of crystalline

units present in the coral skeleton, including both temperature-dependent crystals, COC and fibres also

light-dependent crystals. By considering Fig.7b, only constants from New Caledonia calibration show

slope (a) values higher than –0.19, the slope value derived from the theoretical $\delta^{18}O$–

temperature relationship at equilibrium (Kim et al., 2007), corresponding to coral fragments where





fibres are in higher numbers than COCs, whereas the other slope values lower than –0.19 correspond
to coral portions where COC numbers are higher than fibres.
Considering seasons over a year, an increase (decrease) of temperature induces $\delta^{18}O$ decrease
(increase), and temperature increase (decrease) induces $\delta^{18}O$ increase (decrease) through
photosynthetic increase (decrease) with increasing (decreasing) temperature (Juillet-Leclerc and
Reynaud, 2010; Juillet-Leclerc et al., 2014).

Therefore, for *Porites*, when absolute value of the slope exceeds the absolute value of the quasi-
equilibrium a = –0.20 (b = 0.46) obtained from WW72 data, the value of (a) corresponds to numbers
of COCs compared to fibres due to high temperature, explaining that coral skeleton $\delta^{18}O$ decreases
when coral linear extension increases (Maier et al., 2004).
At every temperature, for annual or monthly resolution, regardless of external conditions, the
distribution of microstructures creates linear relationships between the constants of calibrations of
*Porites*, in turn causing density fluctuations (Gladfelter, 1982; Lough and Cooper, 2011).

*3.2.3 Role of growth rates*
RX images of the cores measured on New Caledonia have been published (DeLong et al., 2013) where
we can see the presence of clear annual banding. However, intra-annual variations of density cannot
be recognised. Only careful identification of seasonal temperature fluctuations, with the sampling path
reported on the RX image, could provide detailed information. But we know that even when the
seasonal density variation is high (Buigues and Bessat, 2001; Lough and Cooper, 2011; Lough and
Cantin, 2014) we cannot attribute clear seasonality to the density change.
DeLong et al. (2013) stress the importance of the orientation of the growth axis, the corallite
distribution and also the distance between density bands underlined by X-rays, knowing that a pair of
dark and clear layers indicates a year's deposit (Barnes and Lough, 1996). X-rays could provide
information about coral growth rates and the density resulting from the interplay of extension and
calcification rates (Lough and Barnes, 2000; Lough, 2008). The latter author remarks: "Routine





examination of coral growth characteristics in conjunction with geochemical analyses of the same
material can greatly enhance the environmental information obtained from coral archives. It is now
admitted that skeletal density results from the interplay of several factors, especially temperature and
light (Tudhope, 1994; Juillet-Leclerc et al., 2006).
As early as 1982, Gladfelter assumed that linear extension and infilling are two independent growth
rates, an assumption supported by Juillet-Leclerc and Reynaud (2010). The authors demonstrated that
each growth rate is related to preferential deposition of microstructures, COCs ensuring linear
extension and fibres, infilling. Furthermore, geochemical investigations reveal that crystal isotopic
signatures differ (Rollion-Bard et al., 2003; Maier et al., 2004; Blamart et al., 2005; Meibom et al.,
2006; Juillet-Leclerc et al., 2009). COC formation should be related to temperature (Gladfelter, 1984)
and fibre deposit depends on both temperature and light (Juillet-Leclerc et al., 2018). Therefore,
temperature and light changes interplay to determine skeletal isotopic composition.
Sampling conducted as it is described in DeLong et al. (2013) includes both COCs and fibres.
Changes of relative amounts of microstructure as illustrated by X-rays and their respective $\delta^{18}O$ are
determined by their mechanisms of formation, unknown so far (Juillet-Leclerc et al., 2009). Following
isotopic laws, the combination of calcification processes and isotopic fractionation could be expressed
as:
$\quad\quad\quad$ measured $\delta^{18}O = [(x_{COC} \times \delta^{18}O_{COC}) + (x_{fibre} \times \delta^{18}O_{fibre})] / (x_{COC} + x_{fibre})$ $\quad\quad$ Eq. (8)
where $x_{COC}$ and $x_{fibre}$ are the relative amounts of the crystal microstructures, with $x_{COC} + x_{fibre} = 1$, and
$\delta^{18}O_{COC}$ and $\delta^{18}O_{fibre}$ are their isotopic signatures depending on temperature and temperature and light,
respectively. This expression is likely to be simplistic but closer to the truth than the thermodynamic
formula. Temperature is the prominent factor because included both in the crystal amounts and the
isotopic signatures.
$SST_{intersection}$ and the corresponding $\delta^{18}O_{intersection}$ should be related to morphology (Land et al., 1975).
When using relationship (8), measured $\delta^{18}O = (x_{COC} \times \delta^{18}O_{COC}) + (x_{fibre} \times \delta^{18}O_{fibre})$, the intersection of
calibration should be obtained when $\delta^{18}O_{intersection} = (0.50 \times \delta^{18}O_{COC}) + (0.50 \times \delta^{18}O_{fiber})$ or at
$SST_{intersection}$, $\delta^{18}O_{intersection} = (\delta^{18}O_{COC} + \delta^{18}O_{fibre})/2$. As long as temperature does not reach $SST_{intersection}$





more fibres are formed in the coral skeleton and temperature exceeds $SST_{intersection}$, COC are
progressively prevailing.

The relationship linking constants (a) and (b) of monthly $\delta^{18}O$ and temperature seems to be inherent to
*Porites* calcification. Slope or (a) ranges between –0.14 to –0.93, surrounding –0.19, the slope value
derived from the theoretical $\delta^{18}O$–temperature relationship at equilibrium (Kim et al., 2007).
Variability of (a) is essentially due to the opposite isotopic effect of simultaneous temperature and
light occurring during the year. Considerations of coral calibrations established from annual and
monthly $\delta^{18}O$ and temperature, reveal the robustness of temperature dependence on isotopic
composition and also highlight the role of intra-annual aragonite density in $\delta^{18}O$ determination. We
conclude that calibrations cannot be explained by simple thermodynamic calculation but need
information about calcification processes and microstructure (COC and fibre) isotope signatures,
depending on temperature and light.

# 4  $\delta^{18}O$ non-linearity over time
## 4.1 Data
*4.1.1 New Caledonia*
Crowley et al. (1999) highlighted $\delta^{18}O$ non-linearity over time for *Porites* from isotopic data series
measured on a core collected at Phare Amédée (New Caledonia) (Quinn et al., 1998), where cores
were also collected for calibrations calculated by Stephans et al. (2004) (paragraph 3.1.1) (Fig. 4).
Crowley et al. (1999) assessed the seasonal calibration established with four samples per year over 22
years. Then, from this calibration, they predicted temperature variations from 1900 until 1992, which
they compared with 20[th] century GISST2 observed temperatures (Parker et al., 1995) following the
same resolution. The calibration cannot be validated, predicted temperatures over 1900–1950 being
underestimated against observed temperatures. Crowley et al. (1999) noticed that by using annual
calibration, the temperature prediction shows better agreement than that derived from monthly
calibration.




*4.1.2 Moorea (French Polynesia)*
We provided another example, using isotopic data measured on a *Porites* core harvested in Moorea
(French Polynesia) (17° 30' S, 149° 50' W) (Boiseau et al., 1998). Fig. 8 illustrates the $\delta^{18}$O non-
linearity in time. On the left side, (Fig. 8a), seasonal measured data are compared with instrumental
seawater temperature between 1980 and 1990 (Boiseau et al., 1998). On the right side, (Fig. 8b), over
the last century, annual averaged measured data, originated from the same data series than seasonal
data, are compared with estimated temperature in the (1°, 1°) grid containing Moorea (Kaplan et al.,
1998). The two curves are displayed to obtain the best matching. The isotopic scale of the two isotopic
profiles is common, while measured and estimated temperature scales cover 7 °C and 2 °C
respectively. There is a mismatch between annual and monthly calibrations given on a unique isotopic
scale.

Evidence underlined by the New Caledonia and Moorea examples must be considered following our
new understanding about environmental forcing.

**4.2 Comparison of annual and monthly $\delta^{18}$O profiles**
The comparison between $\delta^{18}$O profiles and GISST (Parker et al., 1995) or Kaplan (Kaplan et al., 1998)
data sets, derived from statistical assessments, was performed over the last century. Kaplan et al.
(1998) compared ship-derived monthly temperature with the coral-based proxy record from Tarawa
atoll (Cole et al., 1993). The authors observed great discrepancy between the two curves, coral
estimates being difficult to justify.

*4.2.1 Discrepancy between statistical and coral-derived temperature reconstruction*
4.2.1.1 Comparison of annual and monthly calibrations in New Caledonia
We previously displayed monthly calibrations established in New Caledonia (3.2.1 and 3.2.2). Slopes
(a) calculated by Stephans et al. (2004) (Table 2) are similar or higher in absolute value than the slope





from $\delta^{18}$O-temperature calibration utilised by Crowley et al. (1999). The slopes (a) of monthly $\delta^{18}$O–
monthly temperature calibrations (Table 2) are strongly affected by reduced summer isotopic values
corresponding to the highest temperatures, due to the light effect superimposed on the temperature
effect. In contrast, the mean annual isotopic value is not affected by light because this factor varies
weakly over successive years and in turn $\delta^{18}$O essentially reflects temperature. Consequently, when
monthly calibration is applied to predict temperature, the seasonal $\delta^{18}$O being strongly affected by
light induces negative temperature calculations, confirming the effect assessed by Crowley et al.,
(1999). As mentioned in Crowley et al. (1999), when the annual calibration is taken into account, the
prediction of temperature over several decades becomes realistic.

4.2.1.2 Comparison of annual and monthly calibrations in Moorea
We are aware that in Fig. 8, we compare two reconstructions based on different tools. However,
trusting our previous conclusions that annual or monthly $\delta^{18}$O is, to a first approximation, a good
temperature tracer, Fig. 8 illustrates the inconsistency between seasonal and interannual isotopic data.

In Fig. 9a, monthly calibration has been calculated from composite signals over nine years (from1980
to 1989). Since they derive from composite data, calibration constants (a) and (b) may not be
compared with constants from previous relationships (Fig. 7b) (Table 2). In Moorea, where mean
annual temperature is 26.8°C, value of slope (a) from monthly calibration (Fig. 9a) is of the same
order as that in New Caledonia (Crowley et al., 1999). However, the slope value –0.24 derived from
the annual calibration calculated over 33 years (from 1989 to 1956) (Fig. 9b) is lower than the slope
calculated by Crowley et al. (1999), of –0.19. At these sites, rainfall and in turn nebulosity is higher in
November–January, the period recording maximal potential irradiation and maximal temperature
(Boiseau et al., 1998). Despite nebulosity, irradiation affects photosynthetic activity of zooxanthellae
(coral symbionts), strengthened by temperature. Since temperature and light have opposite influences
on $\delta^{18}$O, the slope of monthly calibrations is reduced. Seasonality strongly influences intra-annual or
seasonal isotopic profiles.



During the last century, annual irradiation remained roughly constant. However, global warming
caused by progressive temperature increase is limited around the tropical belt compared to higher
latitudes; however this concept remains a matter of debate (Vecchi and Soden, 2007; Du and Xie,
2009; Zhu and Liu, 2009; Deser et al., 2010). Knowing that the weak temperature increase slightly
impacts photosynthetic activity (Juillet-Leclerc et al., 2014), the single temperature effect on $\delta^{18}O$ is
weakly lower than the calculated effect neglecting light.
The estimation of warming during the 20[th] century deduced from coral monthly calibration, is
estimated to be 1.2 °C (Boiseau et al., 1998), which is too high for a site located in a tropical zone,
whereas the trend of annual $\delta^{18}O$ corresponding to 0.25 °C derived by Kaplan et al. (1998) seems
more realistic.

When coral $\delta^{18}O$ is analysed seasonally, the isotopic profile shows a strong light effect during a year
while two successive years globally do not reflect light change and only weak temperature influence.
Therefore, interannual and monthly $\delta^{18}O$–temperature calibrations for *Porites*, at Moorea and Amédée
Lighthouse are not linear. It is misleading to plot on the time scale monthly $\delta^{18}O$ superimposed on
interannual $\delta^{18}O$ because, both in French Polynesia and New Caledonia, seasonal $\delta^{18}O$ variations are
strongly impacted by both temperature and light and annual variability is slightly influenced by light
and only temperature dependent. Consequently, the global warming of the 20[th] century has to be
estimated from the annual temperature scale to remain realistic.

**5 Consequences for temperature reconstructions**
From the literature dedicated to coral reconstruction based on geochemistry, several papers highlight
the misfit between instrumental temperature and $\delta^{18}O$ (Quinn et al., 2006) and between instrumental
temperature and Sr/Ca records (Alibert and McCulloch, 1997; Crowley et al., 1999, 2000; Nurhati et
al., 2011). Estimates of global warming during the last century as deduced by temperature
reconstructions seem too high for the tropical zone (Damassa et al., 2006; Gorman et al., 2012;
Thierney et al., 2015). A mismatch between seasonal and annual records has been recognized without



any explanation proposed (Osborn et al., 2013; Abram et al., 2015). The possible influence of cloud
cover on proxies is suspected in a few publications (Cahyarini et al., 2014) but is it attributed to
precipitation or to weak photosynthesis?
From previous evidence arise three main concerns: $\delta^{18}O$ is essentially dependent on temperature
according the relationships that include the lack of consistency between the influence of light on
monthly and annual calibrations. All reconstructions based on thermodynamic relationships, such as
the coupled Sr/Ca–$\delta^{18}O$ method or the concept of pseudo-proxy induces biased conclusions. In
addition, confusion between seasonal and annual calibrations causes misleading interpretations.

**5.1 The coupled Sr/Ca–$\delta^{18}O$ method**
Pioneered investigations (McCulloh et al., 1994; Gagan et al., 1998, 2000), Ren et al. (2002) proposed
to deconvolve $\delta^{18}O_{seawater}$ by using subseasonal coral $\delta^{18}O$ and Sr/Ca. This treatment is based on the
oxygen thermometer (3):
$$\delta^{18}O_{carbonate} - \delta^{18}O_{Seawater} = \alpha + \beta \times SST \ (°C) \qquad \text{Eq. (3)}$$
$\alpha$ and $\beta$ being constants, and the Sr/Ca temperature tracer following also linear relationships. The
preliminary condition for applying the Ren et al. (2002) method, "Sr/Ca is solely a function of SST"
may prevent any estimation (Gischler et al., 2005) or it is not respected (Wu et al., 2013), inducing
spurious interpretations. Temperature values are derived from Sr/Ca and $\delta^{18}O$ calibrations assessed
locally from the recent period (Mishima et al., 2010; Cahyarini et al., 2016) or from calibrations
already published (Quinn et al., 2006; Nurhati et al., 2009). Then, this STT value is introduced into
Eq. (3) and $\delta^{18}O_{seawater}$ time series is estimated. This value may be converted into seasurface salinity
(SSS) (Felis et al., 2009; Nurhati et al., 2011; Cahyarini et al., 2014).
The reliability of this method is discussed for multiple reasons: i) we clearly demonstrate that Eq. (3)
does not include light effect, which causes vital effect; ii) Sr/Ca calibration meaning is increasingly
matter of debate (Alibert and Kinsley, 2008; Cahyarini et al., 2008; Alpert et al., 2014), and cultures
testing influence of light on *Acropora* proxies show that reliable Sr/Ca response should be obtained
only under high light intensity (Juillet-Leclerc et al., 2014); iii) when the Sr/Ca and $\delta^{18}O$ temperature




calibrations are applied over a long time scale (more than one century), SST conditions may change at
times, making use of the method difficult (Linsley et al., 2004, 2006). Conversion of $\delta^{18}O_{seawater}$ into
SSS is not always possible, or oceanic advection could confuse SSS reconstruction because the
relationship between $\delta^{18}O_{seawater}$ and SSS is not sufficiently constrained because it is not locally
estimated (Cahyarini et al., 2014; Quinn et al., 2006; Iijima et al., 2005) or may be biased by an
advection (Delcroix et al., 2011).

Climate reconstructions based on the coupled Sr/Ca–$\delta^{18}O$ method must be considered with a critical
eye because of the constraining conditions.

**5.2 Monthly and interannual calibrations**

The consequences of the non-linearity between monthly and annual isotopic data are multiple. The
first implication is the impossibility of plotting a monthly $\delta^{18}O$ curve over several decades or centuries
following an isotopic scale on one side and a temperature scale on the other (Quinn et al., 1998; Cobb
et al., 2003; Abram et al., 2015). A temperature scale deriving from monthly calibration is strongly
impacted by the light effect, whereas an isotopic profile based on annual variability is weakly affected
by light. Therefore, monthly and interannual calibrations, established from a single data series, exhibit
different slopes and monthly and interannual isotopic signals cannot be superimposed. Consequently,
global warming recorded over the 20th century has to be quantified following the annual calibration.
The warming effect assessed from monthly calibration is always overestimated in terms of
temperature (Linsley et al., 2000; Damassa et al., 2006; Tierney et al., 2015).
Oceanographers are commonly face to such a concern as with salinity change not occurring on intra-
annual timescale but noticeable on interannual one. Thus, they commonly use 25 month Hanning filter
to extract the real salinity variability (Gouriou and Delcroix, 2002). From monthly $\delta^{18}O$ profile
covering the total coral core record, it is possible to obtain interannual variability by assessing annual
isotopic averages or by applying 25 month filter. The sole seasonal isotopic record is then calculated




by removing the interannual variations. After obtaining two time series, it is necessary to statistically
treat each of them.

### 5.3 The pseudo-coral concept

First suggested by Thomson et al. (2011), the relationship $\delta^{18}O_{pseudocoral} = a_1$ x SST $+ a_2$ x SSS, $a_1$ and $a_2$
being constants, established for monthly data, presents the advantage of being easily introduced into a
GCM model (Linsley et al., 2017). This could be a good tool to simulate coral isotopic proxy.
However, we have clearly highlighted that $\delta^{18}O_{seawater}$ is included in the coral skeleton $\delta^{18}O$ but
$\delta^{18}O_{seawater}$ and SSS are not always linearly related during times such as in the case of seawater
advection (Delcroix et al., 2011; Linsley et al., 2017). Knowing that the coral $\delta^{18}O$ and temperature
relationship is not linear, it is difficult to include the pseudo-coral concept in paleo-climatic studies
(Gorman et al., 2012; Hereid et al., 2013; Osborn et al., 2013), the latter author noticing 'a mismatch
between seasonal and interannual timescales'. The concept of pseudo-coral is abundantly developed in
terms of theoretical reconstruction techniques (Emile-Geay et al., 2013a, 2013b; Wang et al., 2014).
In order to remedy this deficiency existing in most paleo-climatic studies, Emile-Geay and Tingley
(2015) proposed the use of a simple empirical transform (ITS). However, a much more efficient tool is
the identification of the cause of the non-linearity. Such behaviour has been already highlighted (Felis
et al., 2000; Zhang et al., 2009; Osborne et al., 2013; Abram et al., 2015; Zinke et al., 2014).

### 5.4 Reconstruction of interannual and interdecadal variations by using SSA (Singular Spectrum Analysis) or MTM (Multi-Taper Method)

Briefly, after capturing high- and low-frequencies present in the proxies or reconstructed
environmental parameters on interannual data sets (mean seasonal cycle removed), SSA decomposes
noisy time series into their dominant variance patterns and MTM determines variance spectra and
coherency (Vautard et al., 1992). Climatic variability so studied is the ENSO event occurring at
interannual time scale and at a lower frequency, ITCZ (Inter-Tropical Convergence Zone) migrations,
the PDO (Pacific Decadal Oscillation) or the IOD (Indian Ocean Dipole) and Asian monsoon.



When this method is applied to $\delta^{18}$O profile, it is difficult to separate temperature and/or salinity
change (Felis et al., 2000; Osborne et al., 2014; Cahyarini et al., 2014; Linsley et al., 2017) or to
estimate the interaction between IOD and Indian monsoon (Abram et al., 2008). The use of
sophisticated statistics does not always allow atmospheric and meteorological interactions to be
established if real proxy significance is not considered.

All the methods or concepts highlighted are used abundantly in papers dealing with the reconstruction
of the climate context from coral geochemical tracers; however, they do not respond to the constraints
we have underlined during our demonstration.

**6 Conclusions**
By revisiting several published papers we have highlighted the role of light in $\delta^{18}$O determination,
light so far being an ignored vital effect. Since temperature and light influences are opposite on $\delta^{18}$O,
it is easier to neglect light; however, this explains why synchronous $\delta^{18}$O variability observed in
distinct cores, even synchronous $\delta^{18}$O variability recorded on the same colony, may differ each other.
The WW72 data series reveal that the annual averaged measure of oxygen isotopic ratios performed
on several coral colonies of a single genus, collected at one site, allow comparison due to homogenous
light effects. This allows stronger conclusions. Interpreted with new eyes, we concluded that it is
likely that all coral genera $\delta^{18}$O levels are strongly temperature-dependent and should be used as
tracers of environmental parameters.
Temperature appears to be the dominant factor in $\delta^{18}$O levels because it is recorded in two ways: as a
thermodynamic forcing causing $\delta^{18}$O decrease, and as responsible for photosynthesis enhancement
inducing $\delta^{18}$O increase.
After observing the relationships linking the constants of annual (Sr/Ca)–temperature calibrations
compared to the relationships linking the constants of annual $\delta^{18}$O–temperature calibrations, we
deduced that the analogy should be due to relative amounts of two mineral microstructures, COCs and
fibres. COCs probably depend only on temperature and fibres depend on both light and temperature.



Similar conclusions derive from revisited monthly $\delta^{18}$O–temperature calibrations assessed for *Porites*
coral, in three sites characterised by different annual temperatures. We established a robust
relationship linking the constants of the respective $\delta^{18}$O–temperature calibrations calculated on
multiple *Porites* colonies of different species. Taking into account all *Porites* $\delta^{18}$O–temperature
calibration constants, the high correlation coefficient obtained is at least 0.99, underlining the
consistency of the calibrations. In addition, this indicates the prominent role of temperature in $\delta^{18}$O
levels, acting both thermodynamically and through photosynthetic activity impacted by temperature.
We stress that the relative numbers of mineral microstructures also support this argument.
We explained how light impact differs according to annual or monthly time scales. Annual $\delta^{18}$O
variations are weakly affected by annual light change while monthly variations are strongly affected
by seasonal light. Consequently, a $\delta^{18}$O profile derived from monthly resolution results from the
superposition of annual variations weakly affected by annual light change and monthly variations
strongly impacted by seasonal light fluctuations. When oxygen isotopes are plotted against
temperature, the confusion of time scales generates major misleading. For example, global warming
recorded over the 20$^{\text{th}}$ century derived from a monthly $\delta^{18}$O profile is overestimated.



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





**Table Captions**

**Table 1** – Groups of coral genera from WW72, identified as showing $\delta^{18}$O–temperature calibration
constants linearly linked with correlation coefficient $R^2 \geq 0.99$ (Fig. 3). They have been first
highlighted by calibrations forming bundle characterized by intersections defining $\delta^{18}$O and
temperature ranges (Fig. 1d).

**Table 2** – Values of constant of $\delta^{18}$O–temperature calibrations from WW72, and from New
Caledonia (Stephans et al., 2004), Clipperon (Linsley et al., 1999) and Indonesia (Maier et al., 2004).






**Figure Captions**

**Figure 1** – Figures of the revisited Weber and Woodhead (1972) data series. **Fig. 1a** is the location of
all the islands considered by the authors determining temperatures. **Fig. 1b** displays calibrations
annual temperature–annual $\delta^{18}O$, plotted by considering annual temperature as the unknown
parameters. **Fig. 1c** displays annual $\delta^{18}O$–annual temperature calibrations plotted with annual coral
$\delta^{18}O$ as the unknown parameter, annual temperature being the robust parameter. **Fig. 1d** displays
bundles of annual $\delta^{18}O$–annual temperature calibrations as we identify them from Fig. 1c, for the
groups including *Porites* and *Acropora*. From Fig. 1d, it is possible to generate annual temperature
and annual $\delta^{18}O$ ranges corresponding to the intersection of the calibrations. This feature is made
possible by the homogenous light influence on calibrations.

**Figure 2** – Annual ($\delta^{18}O_{coral}$ - $\delta^{18}O_{seawater}$)-annual temperature calibrations. Weber and Woohead
(1972) data series provided coral data. $\delta^{18}O_{seawater}$ are introduced in the annual $\delta^{18}O$-annual temperature
calibrations according to Juillet-Leclerc and Schmidt (2001) method. Some genera are not present in
all the sites, in turn at all temperatures and only corresponding $\delta^{18}O_{seawater}$ are introduced.
*Acropora*    $\delta^{18}O_{carbonate}$ - $\delta^{18}O_{Seawater}$ = –0.21 x SST (°C) + 1.26, $R^2$ = 0.87, n = 24, p<0.001
*Porites*    $\delta^{18}O_{carbonate}$ - $\delta^{18}O_{Seawater}$ = –0.20 x SST (°C) + 0.45, $R^2$ = 0.83, n = 22, p<0.001
*Montipora*    $\delta^{18}O_{carbonate}$ - $\delta^{18}O_{seawater}$ = –0.19 x SST (°C) + 0.64, $R^2$ = 0.64, n = 12, p<0.05
*Platygira*   $\delta^{18}O_{carbonate}$ - $\delta^{18}O_{seawater}$ = –0.19 x SST (°C) – 0.08, $R^2$ = 0.93, n = 11, p<0.001
*Pavona*    $\delta^{18}O_{carbonate}$ - $\delta^{18}O_{seawater}$ = –0.17 x SST (°C) – 0.47, $R^2$ = 0.87, n = 8, p<0.01
Discrepancies between the different genera calibrations are related to microstructure distribution
characterizing each morphology.

**Figure 3** – Linear relationship between (b) and (a), constants of the annual $\delta^{18}O$–annual temperature
calibrations, $\delta^{18}O_{carbonate}$ = a x SST (°C) + b. Weber and Woohead (1972) data series provided coral





data. **Fig. 3a** displays constants values from the 44 coral genera of Table 1. (a) is considered as the
disequilibrium indicator compared to –0.19, the slope value derived from the theoretical $\delta^{18}O$–
temperature relationship at equilibrium (Kim et al., 2007). The relationship $b = -29.07 \times a - 5.13$, $R^2 =$
0.95, $n = 44$, $p<0.001$ (the green line) takes into account all the data (dark green diamonds), whereas $b$
$= -27.94 \times a - 4.84$, $R^2 = 0.90$, $n = 40$, $p<0.001$ (the blue line) is assessed without the 4 extreme data
(the remaining data are the blue crosses). On **Fig. 3b**, the dots are similar to the dots displayed on Fig.
3a, however, color of the dots corresponds to the color of the calibration bundles of Fig. 1c.
Group I        $b = -24.43 \times a - 4.18$, $R^2 = 0.99$, $n = 9$, $p<0.001$ (the orange line)
Group II        $b = -26.63 \times a - 4.91$, $R^2 = 0.99$, $n = 8$, $p<0.001$ (the violin line)
Group III       $b = -25.85 \times a - 4.10$, $R^2 = 0.99$, $n = 7$, $p<0.001$ (the blue line)
*Acropora* Group IV    $b = -25.60 \times a - 3.79$, $R^2 = 0.99$, $n = 10$, $p<0.001$ (the green line)
*Porites* Group V        $b = -28.40 \times a - 5.16$, $R^2 = 0.999$, $n = 9$, $p<0.001$ (the brown line)
$T_{intersection}$ and $\delta_{intersection}$ are only given for *Acropora* and *Porites* groups.
Correlation coefficient of all the linear relationships are very high. All genera included in each group
share identical microstructure distribution due to common feature of morphology.

**Figure 4** –Graphs derived from Stephans et al. (2004) data, available on NOAA (National Climatic
Data Center site) (https://www.ncdc.noaa.gov/paleo/study/1877). On **Fig. 4a** are reported seasonal
isotopic profiles from 1967 to 1993 period for 92PAC coral core (blue curve), 92PAD coral core (pink
curve), 99PAA coral core (green curve) and 92PAA coral core (violin curve). All the cores have been
harvested at Fort Amédée lighthouse proximity. Seasonal isotopic profiles are strongly impacted by
seasonality with different light influence. **Fig. 4b** displays seasonal $\delta^{18}O$-seasonal temperature (GISS
SST) calibrations for the coral cores studied.
92PAC        $\delta^{18}O_{carbonate} = -0.17 \times SST\ (°C) - 0.08$, $R^2 = 0.77$, $n = 296$, $p<0.001$, blue curve
99PAA        $\delta^{18}O_{carbonate} = -0.16 \times SST\ (°C) - 0.39$, $R^2 = 0.67$, $n = 296$, $p<0.001$, green curve
92PAC        $\delta^{18}O_{carbonate} = -0.15 \times SST\ (°C) - 0.62$, $R^2 = 0.62$, $n = 296$, $p<0.001$, violin curve
92PAD        $\delta^{18}O_{carbonate} = -0.14 \times SST\ (°C) - 1.09$, $R^2 = 0.59$, $n = 296$, $p<0.001$, pink curve



All (a) are higher than –0.19, the slope value derived from the theoretical $\delta^{18}$O–

temperature relationship at equilibrium (Kim et al., 2007). These values indicate that fibers are the

prevailing microstructures of the corals considered.

**Fig. 4c** displays constant (a) and (b) relationship b = –32.6 x a – 5.6, $R^2$ = 0.98, n = 4, p<0.01.

**Figure 5** – Clipperton $\delta^{18}$O data covering the period 1985–1994 (Linsley et al., 1999, 2000),

available on https://www.ncdc.noaa.gov/paleo/study/1846. Three cores are considered 2B, 3C and 4B.

**Fig. 5a** displays $\delta^{18}$O profiles characterized by strong annual variability, 2B (orange curve), 3C (green

curve), and 4B (blue curve). **Fig. 5b** shows the three core seasonal $\delta^{18}$O–monthly temperature

calibrations.

3C        $\delta^{18}O_{carbonate}$ = –0.39 x SST (°C) + 5.26, trend graph derived from 3 temperatures, orange curve

3C        $\delta^{18}O_{carbonate}$ = –0.46 x SST (°C) + 7.4, trend graph derived from 3 temperatures, green curve

4B        $\delta^{18}O_{carbonate}$ = –0.53 x SST (°C) + 9.21, trend graph derived from 3 temperatures, blue curve

The slope values (a) being lower than –0.19, the slope value derived from the theoretical $\delta^{18}$O–

temperature relationship at equilibrium (Kim et al., 2007), correspond to coral colonies grown at high

temperature showing great amount of COC compared to fiber amount.

**Fig. 5c** displays constant (a) and (b) relationship b = –28.21 x a + 20.27, $R^2$ = 0.997, n = 3, p<0.01

**Figure 6** – 6 coral heads representing 3 *Porites* species (*Porites lutea*, *Porites murrayensis* and

*Porites australiensis*), collected in Taka Bone Rate (Indonesia), have been sampled. Each species,

composed by two coral heads, provides four sampling profiles covering 4 years. Each trajectory

presents different light incidence. **Fig.6a** shows all the calibrations. Except one calibration of *Porites*

*australiansis*, all the other calibrations exhibit intersection close to the temperature and $\delta^{18}$O ranges

defined for *Porites* group (Fig. 1d). All the calibrations constants are reported on **Fig. 6b**.

The negative values (a), associated to high linear extension are characteristic features of coral skeleton

grown at high temperature richer in COC than fibres. The correlation coefficient given for all *Porites*

species is high: b = –28.34 x a – 5.59, $R^2$ = 0.999, n = 12, p<0.001




**Figure 7** – **Fig. 7a** displays *Porites* seasonal $\delta^{18}$O–monthly temperature calibrations of New
Caledonia corals (Quinn and Sampson, 2002; Stephans et al., 2004), Clipperton corals (Linsey et al.,
1999, 2000), Taka Bone Rate corals (Maier et al., 2004) and annual $\delta^{18}$O–annual temperature
calibration derived from Weber and Woodhead (1972) data series. On **Fig. 7b** are plotted all the (a)
and (b) values corresponding to the calibrations reported on Fig. 7a. The correlation coefficient given
for all *Porites* species is high: b = –27.24 x a – 4.92, $R^2$ = 0.999, n = 30, p<0.001. All dots showing (a)
> –0.19, the slope value derived from the theoretical $\delta^{18}$O–temperature relationship at equilibrium
(Kim et al., 2007) correspond to New Caledonia coral cores developed at mitigated temperatures, with
fibers in greater amounts compared to COC, all other ones showing (a) < –0.19 are associated to corals
grown at high temperature, with reverse microstructures relative amounts.

**Figure 8** – Comparison of $\delta^{18}$O measured on coral core collected at Moorea (French Polynesia)
(Boiseau et al., 1998) and measured and estimated temperatures. On the left side **Fig. 8a**, between
1980 and 1990, the seasonal measured data are compared to the instrumental seawater temperature
(Boiseau et al., 1998). On the right side **Fig. 8b**, over the last century, the annual averaged measured
data, originated from the same data series than seasonal data, are compared to the temperature
estimated in the (1°, 1°) grid containing Moorea (Kaplan et al., 1998). The two curves are displayed to
obtain the best matching. The isotopic scale of the two isotopic profiles is common to the two profiles,
while the measured and the estimated temperature scales cover 7°C and 2°C respectively. There is a
mismatch between the annual and monthly calibrations given on a unique isotopic scale, illustrating
the non-linearity between the monthly and annual $\delta^{18}$O profiles over the time.

**Figure 9** – Comparison of the monthly composite $\delta^{18}$O–monthly composite temperature calibration
calculated over 1979 to 1989 (Fig. 9a) and the annual $\delta^{18}$O–annual temperature calibration calculated
over 33 years (from 1989 to 1956) (Fig. 9b) (Boiseau et al., 1998). The averaged temperature
calculated from the composite temperature is 25.88 °C whereas the averaged temperature from the last





30 years is 26.7 °C. (a) of the monthly composite $\delta^{18}$O–monthly composite temperature calibration
shown on **Fig. 9a** is –0.15 similar with slope obtained from New Caledonia, however, the composite
temperatures may not be really compared with the measurements. **Fig. 9b** displays the annual $\delta^{18}$O–
annual temperature calibration with the slope (a) slightly lower than –0.19 the slope value derived
from the theoretical $\delta^{18}$O–temperature relationship at equilibrium (Kim et al., 2007) in good
agreement with the values reported on Fig. 7b.






| Genus | Family | Suborder | Group | δ¹⁸O and temperature ranges | | R=Specimen nb/Site nb | b | a |
|---|---|---|---|---|---|---|---|---|
| | | | | SST °C | δ¹⁸O‰ vs VPDB | | | |
| *Platygyra* | Faviidae | FA | | | | 8.23 | 2.24 | -0.27 |
| *Leptoria* | Faviidae | FA | | | | 3.36 | 2.37 | -0.27 |
| *Goniopora* | Poritidae | FU | | | | 6.46 | 0.99 | -0.21 |
| *Goniastrea* | Faviidae | FA | I | 24.42 | -4.18 | 7.00 | 1.72 | -0.24 |
| *Echinophyllia* | Faviidae | FA | | | | 2.14 | 2.29 | -0.26 |
| *Oxypora* | Pectiniidae | FA | | | | 1.80 | 1.32 | -0.22 |
| *Astreopora* | Fungiidae | A | | | | 4.67 | 0.72 | -0.20 |
| *Favites* | Faviidae | FA | | | | 6.55 | 1.67 | -0.24 |
| *Plesiastrea* | Faviidae | FA | | | | 6.00 | 1.82 | -0.24 |
| *Coeloseris* | Agariciidae | FU | | | | 4.80 | -0.48 | -0.17 |
| *Caulastrea* | Faviidae | FA | | | | 3.00 | 2.96 | -0.30 |
| *Acrhelia* | Faviidae | FA | | | | 2.50 | 0.44 | -0.20 |
| *Oulophyllia* | Faviidae | FA | II | 26.63 | -4.91 | 2.50 | -1.68 | -0.12 |
| *Lobophyllia* | Mussidae | FA | | | | 7.07 | 0.92 | -0.22 |
| *Symphyllia* | Mussidae | FA | | | | 3.75 | 1.09 | -0.22 |
| *Favia* | Faviidae | FA | | | | 6.56 | 1.53 | -0.24 |
| *Acanthastrea* | Mussidae | FA | | | | 2.30 | 2.72 | -0.28 |
| *Pavona* | Agariciidae | FU | | | | 7.94 | 2.18 | -0.25 |
| *Alveopora* | Poritidae | FU | | | | 3.40 | 1.44 | -0.21 |
| *Diploastrea* | Faviidae | FA | | | | 2.17 | 2.11 | -0.24 |
| *Cyphastrea* | Faviidae | FA | III | 25.85 | -4.10 | 3.81 | 1.08 | -0.20 |
| *Fungia* | Fungiidae | FU | | | | 13.62 | 2.74 | -0.26 |
| *Polyphyllia* | Fungiidae | FU | | | | 2.57 | 1.24 | -0.21 |
| *Leptastrea* | Fungiidae | FU | | | | 5.21 | 2.01 | -0.23 |
| *Pliesioseris* | Thamnastreiidae | A | | | | 3.40 | 2.08 | -0.23 |
| *Psammocora* | Thamnastreiidae | A | | | | 5.87 | 2.03 | -0.23 |
| *Parahalomitra* | Fungiidae | FU | | | | 2.56 | 3.99 | -0.31 |
| *Coscinarea* | Siderastreiidae | FU | | | | 3.43 | 2.85 | -0.26 |
| *Herpolitha* | Fungiidae | FU | | | | 2.22 | 5.39 | -0.36 |
| *Seriatopora* | Pocilloporidae | A | IV | 25.6 | -3.79 | 4.44 | 3.11 | -0.27 |
| *Stephanaria* | Thamnastreiidae | A | | | | 1.89 | 4.10 | -0.31 |
| *Turbinaria* | Dendrophyllidae | D | | | | 6.43 | 4.01 | -0.30 |
| *Montipora* | Acroporidae | A | | | | 11.70 | 3.76 | -0.29 |
| *Acropora* | Acroporidae | A | | | | 30.93 | 3.43 | -0.28 |
| *Stylophora* | Pocilloporidae | A | | | | 6.80 | 2.02 | -0.22 |
| *Euphyllia* | Caryophylliidae | C | | | | 5.11 | 0.60 | -0.20 |
| *Merulina* | Merulinidae | FA | | | | 3.75 | 0.68 | -0.21 |
| *Pectinea* | Pectiniidae | FA | | | | 2.80 | -0.58 | -0.16 |
| *Galaxea* | Oculinidae | FA | | | | 5.07 | 1.98 | -0.25 |
| *Hydnophora* | Faviidae | FA | V | 28.4 | -5.16 | 3.89 | 2.87 | -0.28 |
| *Echinopora* | Faviidae | FA | | | | 5.27 | 2.93 | -0.28 |
| *Porites* | Poritidae | FU | | | | 16.19 | 3.39 | -0.30 |
| *Pocillopora* | Acroporidae | A | | | | 9.33 | 2.37 | -0.26 |
| *Mycedium* | Pectiniidae | FA | | | | 2.00 | 3.87 | -0.32 |

**Table 1** – Groups of coral genera from WW72, identified as showing δ¹⁸O–temperature calibration
constants linearly linked with correlation coefficient $R^2 \geq 0.99$ (Fig. 3). They have been first
highlighted by calibrations forming bundle characterized by intersections defining δ¹⁸O and
temperature ranges (Fig. 1d).






| | $\delta^{18}O = a*SST + b$ | |
|---|---|---|
| | **a** | **b** |
| | -0.20 | 0.60 |
| | -0.21 | 0.68 |
| | -0.16 | -0.58 |
| | -0.25 | 1.98 |
| **from WW72** | -0.28 | 2.87 |
| | -0.28 | 2.93 |
| | -0.30 | 3.39 |
| | -0.26 | 2.37 |
| | -0.32 | 3.87 |
| | -0.17 | -0.08 |
| **from Stephans et al., 2004** | -0.16 | -0.39 |
| | -0.15 | -0.62 |
| | -0.14 | -1.09 |
| | -0.46 | 7.4 |
| **from Linsley et al., 1999** | -0.53 | 9.21 |
| | -0.39 | 5.26 |
| | -0.78 | 16.29 |
| | -0.80 | 17.06 |
| | -0.56 | 10.29 |
| | -0.61 | 11.86 |
| | -0.59 | 11.17 |
| **from Maier et al., 2004** | -0.47 | 7.76 |
| | -0.47 | 7.58 |
| | -0.43 | 6.83 |
| | -0.51 | 8.91 |
| | -0.38 | 5.10 |
| | -0.93 | 20.92 |
| | -0.43 | 6.35 |

**Table 2** – Values of constant of $\delta^{18}O$–temperature calibrations from WW72, and from New
Caledonia (Stephans et al., 2004), Clipperon (Linsley et al., 1999) and Indonesia (Maier et al., 2004).




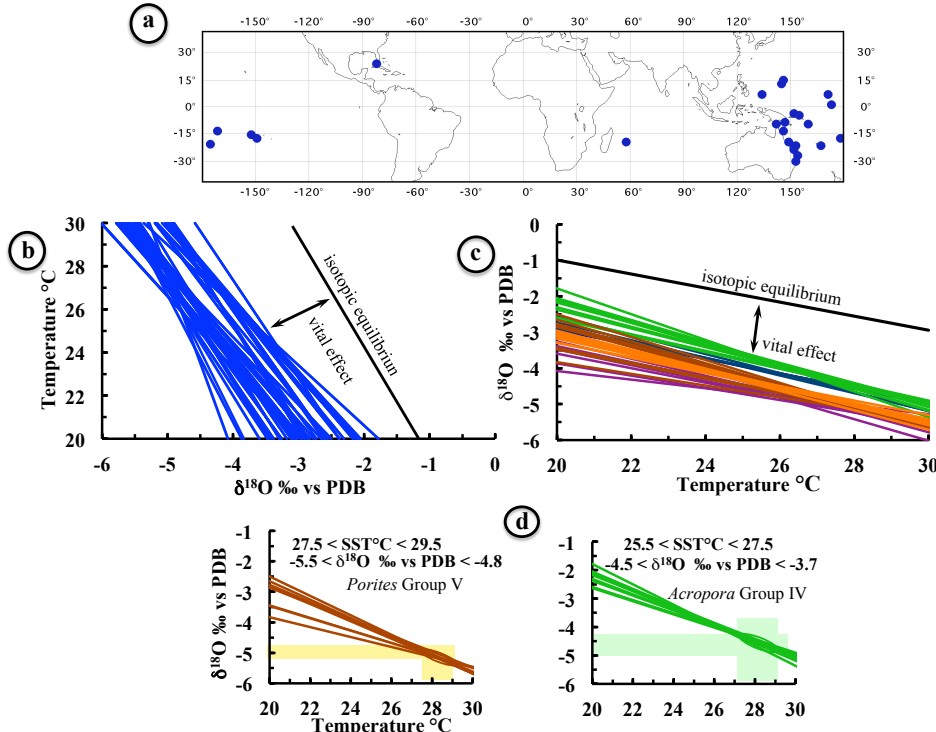

**Figure 1** – Figures of the revisited Weber and Woodhead (1972) data series. **Fig. 1a** is the location of
all the islands considered by the authors determining temperatures. **Fig. 1b** displays calibrations
annual temperature–annual $\delta^{18}$O, plotted by considering annual temperature as the unknown
parameters. **Fig. 1c** displays annual $\delta^{18}$O-annual temperature calibrations plotted with annual coral
$\delta^{18}$O as the unknown parameter, annual temperature being the robust parameter. **Fig. 1d** displays
bundles of annual $\delta^{18}$O-annual temperature calibrations as we identify them from Fig. 1c, for the
groups including *Porites* and *Acropora*. From Fig. 1d, it is possible to generate annual temperature
and annual $\delta^{18}$O ranges corresponding to the intersection of the calibrations. This feature is made
possible by the homogenous light influence on calibrations.




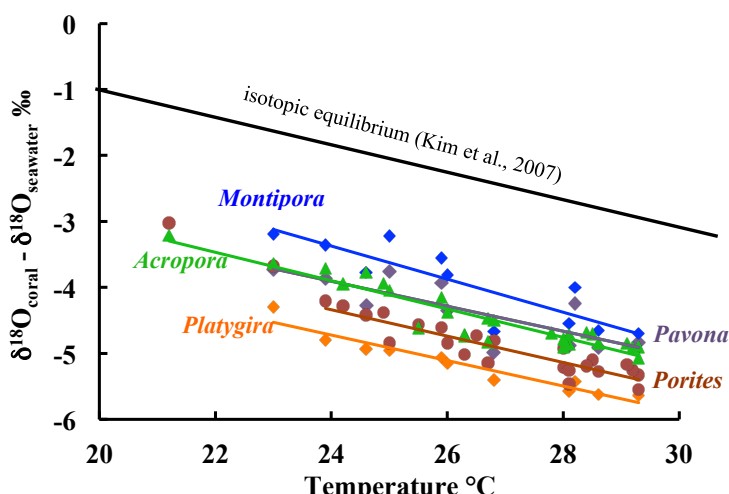

**Figure 2** – Annual ($\delta^{18}O_{coral} - \delta^{18}O_{seawater}$)-annual temperature calibrations. Weber and Woohead
(1972) data series provided coral data. $\delta^{18}O_{seawater}$ are introduced in the annual $\delta^{18}O$-annual temperature
calibrations according to Juillet-Leclerc and Schmidt (2001) method. Some genera are not present in
all the sites, in turn at all temperatures and only corresponding $\delta^{18}O_{seawater}$ are introduced.
*Acropora*   $\delta^{18}O_{carbonate} - \delta^{18}O_{seawater} = -0.21 \times SST\ (°C) + 1.26$, $R^2 = 0.87$, n = 24, p<0.001
*Porites*   $\delta^{18}O_{carbonate} - \delta^{18}O_{seawater} = -0.20 \times SST\ (°C) + 0.45$, $R^2 = 0.83$, n = 22, p<0.001
*Montipora*    $\delta^{18}O_{carbonate} - \delta^{18}O_{seawater} = -0.19 \times SST\ (°C) + 0.64$, $R^2 = 0.64$, n = 12, p<0.05
*Platygira*   $\delta^{18}O_{carbonate} - \delta^{18}O_{seawater} = -0.19 \times SST\ (°C) - 0.08$, $R^2 = 0.93$, n = 11, p<0.001
*Pavona*    $\delta^{18}O_{carbonate} - \delta^{18}O_{seawater} = -0.17 \times SST\ (°C) - 0.47$, $R^2 = 0.87$, n = 8, p<0.01
Discrepancies between the different genera calibrations are related to microstructure distribution
characterizing each morphology.






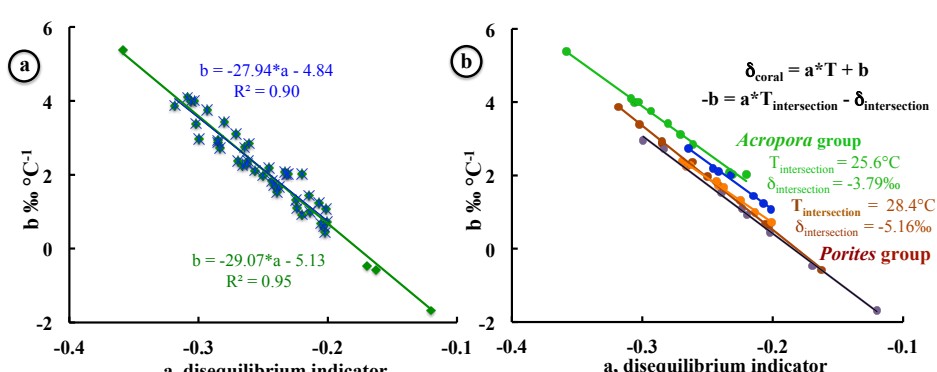

**Figure 3** – Linear relationship between (b) and (a), constants of the annual $\delta^{18}$O-annual temperature
calibrations, $\delta^{18}O_{carbonate}$ = a x SST (°C) + b. Weber and Woohead (1972) data series provided coral
data. **Fig. 3a** displays constants values from the 44 coral genera of Table 1. (a) is considered as the
disequilibrium indicator compared to –0.19, the slope value derived from the theoretical $\delta^{18}$O-
temperature relationship at equilibrium (Kim et al., 2007). The relationship b = –29.07 x a – 5.13, $R^2$ =
0.95, n = 44, p<0.001 (the green line) takes into account all the data (dark green diamonds), whereas b
= –27.94 x a – 4.84, $R^2$ = 0.90, n = 40, p<0.001 (the blue line) is assessed without the 4 extreme data
(the remaining data are the blue crosses). On **Fig. 3b**, the dots are similar to the dots displayed on Fig.
3a, however, color of the dots corresponds to the color of the calibration bundles of Fig. 1c.
Group I        b = –24.43 x a – 4.18, $R^2$ = 0.99, n = 9, p<0.001 (the orange line)
Group II       b = –26.63 x a – 4.91, $R^2$ = 0.99, n = 8, p<0.001 (the violin line)
Group III      b = –25.85 x a – 4.10, $R^2$ = 0.99, n = 7, p<0.001 (the blue line)
*Acropora* Group IV    b = –25.60 x a – 3.79, $R^2$ = 0.99, n = 10, p<0.001 (the green line)
*Porites* Group V        b = –28.40 x a – 5.16, $R^2$ = 0.999, n = 9, p<0.001 (the brown line)
$T_{intersection}$ and $\delta_{intersection}$ are only given for *Acropora* and *Porites* groups.
Correlation coefficient of all the linear relationships are very high. All genera included in each group
share identical microstructure distribution due to common feature of morphology.

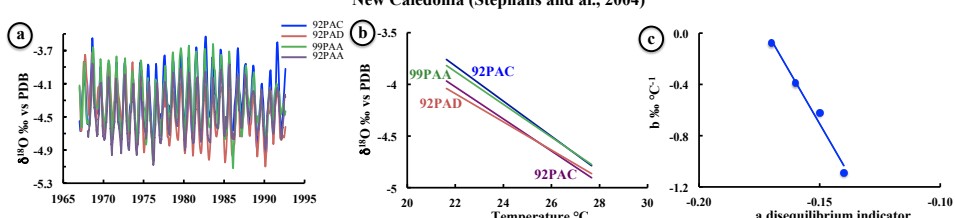

**Figure 4** –Graphs derived from Stephans et al. (2004) data, available on NOAA (National Climatic
Data Center site) (https://www.ncdc.noaa.gov/paleo/study/1877). On **Fig. 4a** are reported seasonal
isotopic profiles from 1967 to 1993 time period for 92PAC coral core (blue curve), 92PAD coral core
(pink curve), 99PAA coral core (green curve) and 92PAA coral core (violin curve). All the cores have
been harvested at Fort Amédée lighthouse proximity. Seasonal isotopic profiles are strongly impacted
by seasonality with different light influence. **Fig. 4b** displays seasonal $\delta^{18}$O–seasonal temperature
(GISS SST) calibrations for the coral cores studied.
92PAC $\delta^{18}O_{carbonate}$ = –0.17 x SST (°C) – 0.08, $R^2$ = 0.77, n = 296, p<0.001, blue curve
99PAA $\delta^{18}O_{carbonate}$ = –0.16 x SST (°C) – 0.39, $R^2$ = 0.67, n = 296, p<0.001, green curve
92PAC $\delta^{18}O_{carbonate}$ = –0.15 x SST (°C) – 0.62, $R^2$ = 0.62, n = 296, p<0.001, violin curve
92PAD $\delta^{18}O_{carbonate}$ = –0.14 x SST (°C) – 1.09, $R^2$ = 0.59, n = 296, p<0.001, pink curve
All (a) are higher than –0.19, the slope value derived from the theoretical $\delta^{18}$O-
temperature relationship at equilibrium (Kim et al., 2007). These values indicate that fibers are the
prevailing microstructures of the corals considered.
**Fig. 4c** displays constant (a) and (b) relationship b = –32.6 x a – 5.6, $R^2$ = 0.98, n = 4, p<0.01.

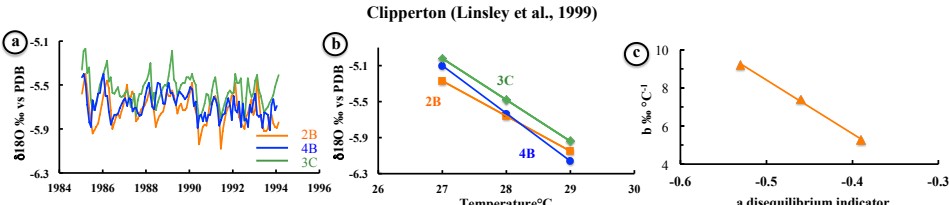

**Figure 5** – Clipperton  $\delta^{18}$O data covering 1985-1994 period (Linsley et al., 1999, 2000), available on
https://www.ncdc.noaa.gov/paleo/study/1846. Three cores are considered 2B, 3C and 4B. **Fig. 5a**
displays $\delta^{18}$O profiles characterized by strong annual variability, 2B (orange curve), 3C (green curve),
and 4B (blue curve). **Fig. 5b** shows the three core seasonal $\delta^{18}$O–monthly temperature calibrations.
3C   $\delta^{18}O_{carbonate}$ = –0.39 x SST (°C) + 5.26, trend graph derived from 3 temperatures, orange curve
3C   $\delta^{18}O_{carbonate}$ = –0.46 x SST (°C) + 7.4, trend graph derived from 3 temperatures, green curve
4B   $\delta^{18}O_{carbonate}$ = –0.53 x SST (°C) + 9.21, trend graph derived from 3 temperatures, blue curve
The slope values (a) being lower than –0.19, the slope value derived from the theoretical $\delta^{18}$O-
temperature relationship at equilibrium (Kim et al., 2007), correspond to coral colonies grown at high
temperature showing great amount of COC compared to fibre amount.
**Fig. 5c** displays constant (a) and (b) relationship b = –28.21 x a + 20.27, $R^2$ = 0.997, n = 3, p<0.01






**Taka Bone Rate (Indonesia) (Maier et al., 2004)**

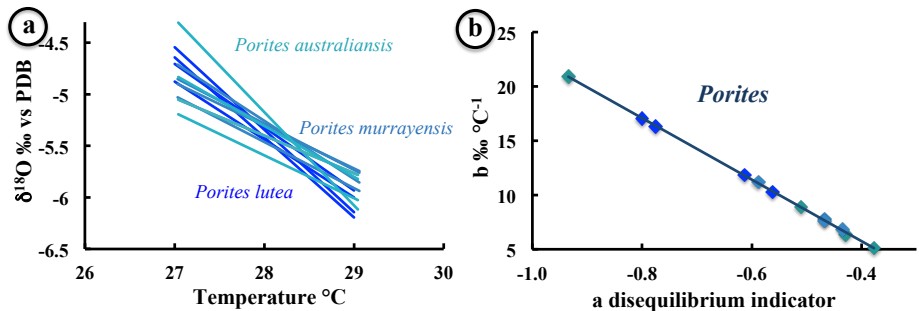

**Figure 6** – 6 coral heads representing 3 *Porites* species (*Porites lutea*, *Porites murrayensis* and
*Porites australiensis*), collected in Taka Bone Rate (Indonesia), have been sampled. Each species,
composed by two coral heads, provides four sampling profiles covering 4 years. Each trajectory
presents different light incidence. **Fig.6a** shows all the calibrations. Except one calibration of *Porites*
*australiansis*, all the other calibrations exhibit intersection close to the temperature and $\delta^{18}O$ ranges
defined for *Porites* group (Fig. 1d). All the calibrations constants are reported on **Fig. 6b**.
The negative values (a), associated to high linear extension are characteristic features of coral skeleton
grown at high temperature richer in COC than fibres. The correlation coefficient given for all *Porites*
species is high: b = –28.34 x a – 5.59, $R^2$ = 0.999, n = 12, p<0.001




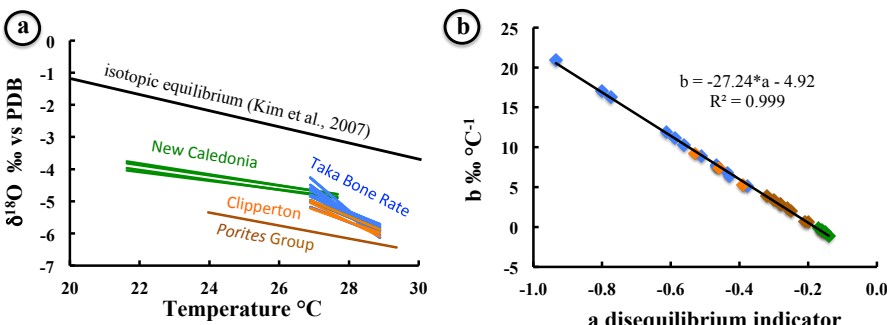

**Figure 7** – **Fig. 7a** displays *Porites* seasonal $\delta^{18}$O-monthly temperature calibrations of New
Caledonia corals (Quinn and Sampson, 2002; Stephans et al., 2004), Clipperton corals (Linsey et al.,
1999, 2000), Taka Bone Rate corals (Maier et al., 2004) and annual $\delta^{18}$O-annual temperature
calibration derived from Weber and Woodhead (1972) data series. On **Fig. 7b** are plotted all the (a)
and (b) values corresponding to the calibrations reported on Fig. 7a. The correlation coefficient given
for all *Porites* species is high: $b = -27.24 \times a - 4.92$, $R^2 = 0.999$, $n = 30$, $p<0.001$. All dots showing (a)
> –0.19, the slope value derived from the theoretical $\delta^{18}$O-temperature relationship at equilibrium
(Kim et al., 2007) correspond to New Caledonia coral cores developed at mitigated temperatures, with
fibers in greater amounts compared to COC, all other ones showing (a) < –0.19 are associated to corals
grown at high temperature, with reverse microstructures relative amounts.




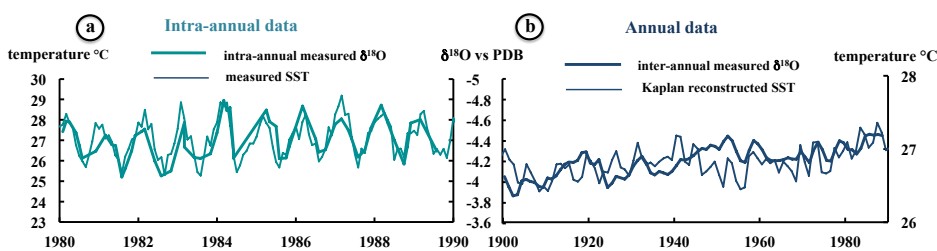

**Figure 8** – Comparison of δ$^{18}$O measured on coral core collected at Moorea (French Polynesia)
(Boiseau et al., 1998) and measured and estimated temperatures. On left hand **Fig. 8a**, between 1980
and 1990, seasonal measured data are compared to instrumental seawater temperature (Boiseau et al.,
1998). On right hand **Fig. 8b**, over the last century, annual averaged measured data, originated from
the same data series than seasonal data, are compared to estimated temperature in the (1°, 1°) grid
containing Moorea (Kaplan et al., 1998). The two curves are displayed to obtain the best matching.
Isotopic scale of the two isotopic profiles is common to the two profiles, while measured and
estimated temperature scales cover 7°C and 2°C respectively. There is a mismatch between annual and
monthly calibrations given on a unique isotopic scale, illustrating the non-linearity between monthly
and annual δ$^{18}$O profiles over the time.




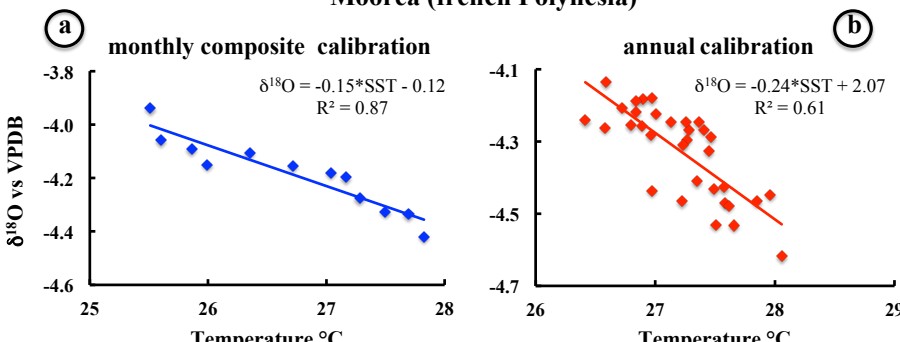

**Figure 9** – Comparison of monthly composite $\delta^{18}$O-monthly composite temperature calibration
calculated over 1979 to 1989 (Fig. 9a) and annual $\delta^{18}$O-annual temperature calibration calculated over
33 years (from 1989 to 1956) (Fig. 9b) (Boiseau et al., 1998). Averaged temperature calculated from
composite temperature is 25.88 °C whereas averaged temperature from the last 30 years is 26.7 °C. (a)
of the monthly composite $\delta^{18}$O-monthly composite temperature calibration shown on **Fig. 9a** is –0.15
similar with slope obtained from New Caledonia, however, composite temperatures may not be really
compared with measurements. **Fig. 9b** displays annual $\delta^{18}$O-annual temperature calibration with slope
(a) slightly lower than –0.19 the slope value derived from the theoretical $\delta^{18}$O-temperature relationship
at equilibrium (Kim et al., 2007) in good agreement with values reported on Fig. 7b.