# Peer review of "The role of light as vital effect on coral skeleton oxygen"

_Biogeosciences, 2018_

## Referee Comment (RC1) · Anonymous Referee #1 · 30 Nov 2018

The manuscript by Juillet-Leclerc addresses the following question:

How does light intensity affect the relationship between a coral oxygen isotope ratio (d18O) and sea surface temperature (SST)?

The author previously reported an increase in coral d18O with increasing light intensity (Reynaud-Vaganay et al., 2001; Juillet-Leclerc and Reynaud, 2010) and now revisits previously published coral d18O data to explore the potential effect of light on the variability of coral d18O-SST relationships.

The problem addressed in this manuscript is of great relevance to the palaeoceanographic community but the method employed by the author prevents any significant and conclusive result. The main methodological issues are detailed below:

1. The author neglects the effect of seawater d18O on the individual coral d18O-SST calibrations discussed in the MS and attributes most of the calibration differences to the effect of light. It is well established that temporal variation in SST and seawater d18O are commonly related (e.g. Cobb et al., 2001), which significantly affects a coral d18O-SST relationship. I strongly recommend carrying the data analysis using 'd18O_coral-d18O_seawater' instead of 'd18O_coral' alone.

2. Most of the demonstration is focused on finding the cause of variations in the slope (a) and intercept (b) of coral d18O-SST relationships. The author uses the slope (a) as an indicator for isotopic disequilibrium: meaning a slope deviating from "-0.2 per mil/degC" indicates isotopic disequilibrium. Again, the effect of seawater d18O on the coral d18O-SST relationship (including the slope) greatly varies with location and cannot be ignored.

3. 'Light intensity' is supposed to be the main parameter tested/discussed in this MS but light intensity data are not shown on any figures or table. The author speculates on a potential link between light and observed coral d18O-SST relationships without any clear evidence of a link between the two parameters.

4. I could not understand why a correlation between the slope and the intercept of the coral d18O-SST relationship had something to do with light intensity and/or coral calcification mechanisms. I do not discard a potential link between these parameters but was simply unable to follow the author reasoning. More generally, I do not think that a link (whether empirical or mechanistic) between light and coral d18O can be assessed with the data presented in this MS.

5. The kinetic isotope effect of McConnaughey (1989) and the role of carbonic anhydrase (Devriendt et al., 2017, Chen et al., 2018) on coral d18O are neglected in this MS.

As a general recommendation, the data compiled in the MS is interesting and could serve another purpose than testing the role of light on coral d18O. A more general

paper on coral annual vs seasonal d18O-SST relationship seems more adapted.

---

## Author Comment (AC1) · 11 Dec 2018

a The referee #1 forgets to mention Juillet-Leclerc et al. (2018), where the light role is highlighted at micrometer size. In all the references the light influence on coral skeleton $\delta$18O is proved. The following manuscript is an up-scaling of the conclusions of Juillet-Leclerc et al. (2018).

b The method that I employ in my manuscript is the listing of some details from coral literature where the lack of light effect induced biases. Of course, I cannot show the light record corresponding to the studies that I referred to but knowing light effect on coral skeleton $\delta$18O, I am able to recognize and explain light impact on oxygen isotope.

1 $\delta$18O-SST In the paragraph 2.1.2, I refer the consequences on the correlation coefficient of the calibration annual $\delta$18O–annual SST after introducing $\delta$18Oseawater. In the paragraph 2.2.1.2, I explain that $\delta$18Oseawater is included in skeleton $\delta$18O but at a lesser degree than SST. Where is light in this comment? In the paragraph 2.2.1.1, I explain that in the term 'temperature', light is hidden as the trigger of photosynthesis increase (decrease) due to temperature increase (decrease) and how $\delta$18O is indirectly affected by light.

2 The slope (a) as an indicator for isotopic disequilibrium Annual Calibrations In the paragraph 2.1.3, I highlight that the constants (a) and (b) from a calibration annual $\delta$18O–annual SST are strongly related. In the paragraph 2.2.2, I consider that a=-0.19 being the theoretical slope of $\delta$18O temperature equation at equilibrium (Kim et al., 2007), the other values of (a) reflect more or less great degree of disequilibrium. The link existing between the constants of annual $\delta$18O–annual SST and those of annual SïĄšïĂŕCïĄą–annual SST allows the link to the relative distribution of microstructures in coral aragonite to be demonstrated. Where is light in this comment? Taking into account Juillet-Leclerc and Reynaud (2010), it is easy to relate fibre existence, one of the aragonite microstructures to light influence on $\delta$18O.

Monthly Calibrations In the paragraph 3.1, I mention all the (a) and (b) relationships deduced from the studies chosen for the demonstration. I consider again, that a=-0.19 being the theoretical slope of $\delta$18O temperature equation at equilibrium (Kim et al., 2007), the other values of (a) reflect more or less great degree of disequilibrium. After explaining the local potential light impact on monthly $\delta$18O in paragraph 3.2.1, the relationship between constant of monthly $\delta$18O-monthly temperature is related to the aragonite microstructures distribution identical to that of annual calibrations, which is recalled on figure captions of Figure 2, Figure 3, Figure 4, Figure 5, Figure 6 and Figure 7. Where is light in this comment? Taking into account Juillet-Leclerc and Reynaud (2010), it is easy to relate fibre existence, one of the aragonite microstructures to light influence on $\delta$18O.

3-4 The factor 'Light intensity' In the introduction of paragraph 2.2, I explain how different light incidences may affect coral growth condition, considerations well known by biologists. In the paragraph 2.2.1, I explain how temperature is recorded twice in $\delta18O$, which is indirectly affected by light. This part of the manuscript justifies the role of light, which strengthens WW72 conclusions. A part of the response is given in b. Other justifications figure in 2.

5 Kinetic process In Juillet-Leclerc et al. (2018), it is demonstrated that coral skeleton $\delta18O$ results of kinetic isotopic fractionation because isotope measures are conducted at micrometer size scale. The conclusion arguments are supported by the biological control of aragonite crystallization. This process is in opposite with Devriendt et al. (2017) and Chen et al. (2018) papers, based on a coral mineralization process purely of physical origin. The kinetic process discussed in Juillet-Leclerc et al. (2018), is not related to calcification rate as is defined by Barnes and Lough (1996) and in Mc-Connaughey (1989) but rather to kinetic fractionations affecting $H_2O$-$CO_2$ system or $CaCO_3$ molecules. The present manuscript does not concern molecular processes and does not involved calcification rates.

Conclusion I admit that when light effect is not identified as soon as the first paragraphs, it is difficult to pay attention before the last paragraphs, comparing annual and seasonal $\delta18O$-SST calibrations.

---

## Short Comment (SC1) · 22 Dec 2018

Thank you for your contribution to the inter-journal special issue 'Paleoclimate data synthesis and analysis of associated uncertainty' and for your help to promote open source paleoclimate science. One of the goals of the special issue on 'Paleoclimate data synthesis and analysis of associated uncertainty' is to promote good data stewardship in paleoclimatology. Therefore, the editors review the data handling of all contributions to the special issue independently from the normal peer reviews and short comments. It will be reviewed whether all data and data handling presented in the submissions are made available freely and adhere to the FAIR concept (https://www.nature.com/articles/sdata201618).

[Figure]

Here are the specific comments for the manuscript of Anne Juillet-Leclerc: (1) Please add a "Data availability statement" in the revised version of the manuscript. This should include at one place the information where the data sets used in this Review paper are available. (2) While links to the other datasets are provided no link exists to the WW72 data set. Please add a link or upload the dataset as appendix to this manuscript. (3) Indicate in details which software and which packages and which formula have been used for the analyses. We suggest that you provide all code so that all results and all figures shown in the publication could be reproduced from original data.

With kind regards, Ulrike Herzschuh on behalf of the guest editorial team